# Conjugation of Mannans to Enhance the Potency of Liposome Nanoparticles for the Delivery of RNA Vaccines

**DOI:** 10.3390/pharmaceutics13020240

**Published:** 2021-02-09

**Authors:** Roshan Goswami, Derek T. O’Hagan, Roberto Adamo, Barbara C. Baudner

**Affiliations:** 1mAbxience, Julia Morros s/n, Armunia, 24009 León, Spain; roshan.goswami30@gmail.com; 2GSK, Via Fiorentina 1, 53100 Siena, Italy; 3GSK, 14200 Shady Grove Road, Rockville, MD 20850, USA

**Keywords:** liposomes, mannosylation, self-amplifying RNA, RSV, vaccines

## Abstract

Recent approval of mRNA vaccines to combat COVID-19 have highlighted the potential of this platform. Lipid nanoparticles (LNP) is the delivery vehicle of choice for mRNA as they prevent its enzymatic degradation by encapsulation. We have recently shown that surface exposition of mannose, incorporated in LNPs as stable cholesterol-amine conjugate, enhances the potency of self-amplifying RNA (SAM) replicon vaccines through augmented uptake by antigen presenting cells (APCs). Here, we generated a new set of LNPs whose surface was modified with mannans of different length (from mono to tetrasaccharide), in order to study the effect on antibody response of model SAM replicon encoding for the respiratory syncytial virus fusion F protein. Furthermore, the impact of the mannosylated liposomal delivery through intradermal as well as intramuscular routes was investigated. The vaccine priming response showed to improve consistently with increase in the chain length of mannoses; however, the booster dose response plateaued above the length of disaccharide. An increase in levels of IgG1 and IgG2a was observed for mannnosylated lipid nanoparticles (MLNPs) as compared to LNPs. This work confirms the potential of mannosylated SAM LNPs for both intramuscular and intradermal delivery, and highlights a disaccharide length as sufficient to ensure improved immunogenicity compared to the un-glycosylated delivery system.

## 1. Introduction

Vaccines save many lives by prevention of diseases and are considered a very cost-effective healthcare intervention. Vaccines are especially relevant in the current COVID-19 pandemic situation, which is one of the most formidable challenges to humanity, impacting health, way of living, and potentially all business activities. Emerging and challenging pathogens require novel approaches to vaccine development and among the emerging approaches, nucleic acid-based vaccines (like viral vectors, DNA, and RNA) have shown the potential of priming of both B and T cell responses by mimicking live infection as they express antigens in situ [1]. In fact, recombinant viral vectored and nucleic acid-based vaccines share almost half to the total experimental vaccines under development and are among the frontrunners in the advanced clinical stages of investigation [2]. The two vaccines obtaining approval in the US for emergency use to prevent COVID-19 are BioNTech’s BNT162 and Moderna’s mRNA1273, both based on the mRNA vaccine platform and with reported efficacy higher than 90% [3,4,5]. The first one recently was approved also in the EU. Apart from offering comparatively higher efficacy, at least in case of COVID-19 vaccines to-date, mRNA vaccines have advantages over viral vectored vaccines in terms of highly reduced potential of genetic transformation and very low propensity to induce anti-vector immunity [6].

As all mRNA vaccines are prone to degradation by nucleases, they suffer poor stability. Many delivery systems have been tested to enhance the stability of such vaccines [7]. For mRNA delivery to be efficient, it must undergo cellular uptake and endosomal escape to express proteins. Cationic lipids and polymers have been explored extensively in this area to trigger cytoplasmic delivery of RNA [8,9,10,11].

However, recently, attention has been focused on Lipid nanoparticles (LNPs) that can encapsulate the nucleic acids, reducing their degradation by nucleases and offering the additional advantages of a lower toxicity and a better transfection efficiency compared to other delivery systems [12]. Furthermore, LNPs be can readily modified to target specific subsets of antigen-presenting cells (APCs), which makes them more suitable for vaccine candidates [13,14]. Interestingly, both mRNA1273 and BNT162 have adopted LNP (lipid nanoparticles) as a delivery system of their mRNA payload in clinical trials for the COVID-19 vaccine [15].

Muscle cells have been shown to be involved in presentation of RNA and DNA vector encoded antigens have been suggested [16,17]. Protein expression seems to occur mostly in muscle somatic cells upon intramuscular injection of naked or liposome delivered mRNA, as well as self-amplifying RNA (SAM) vaccines [16,18,19]. Additionally, the cross presentation by APCs of myocyte-derived antigens has been demonstrated to be the primary mechanism for priming CD8 T cells [20,21]. All this evidence points to the potential of enhancing T-cell-mediated immune response induced by mRNA vaccines by targeted delivery to APCs [1,6].

Mannose (Man) receptors are highly abundant on the cell surface of APCs [13,22,23]. There are few examples of mannan conjugate ligands which have been tested as immunomodulators and vaccine adjuvants [24,25]. DC-SIGN receptor is present especially on dermal and mucosal immature conventional DCs, whereas langerin receptor is mostly expressed on skin epidermal and all stratified epithelial DC subset known as Langerhans cells (LC). Carbohydrate-recognition domain (CRD) on such calcium-dependent C-type lectin receptors of DC-SIGN and langerin are optimal for targeting using mannosylated-LNP for the delivery of SAM vaccines through the intradermal route instead of conventional intramuscular or subcutaneous routes for vaccines [23,26]. However, this approach can benefit targeting the intramuscular DC-SIGN receptor as well, to a considerable extent, owing to the presence of these receptors on immature DCs in muscle [27].

APC targeting through mannose receptors has been already explored for targeted DNA and RNA-based antitumor immunotherapy [28] and in the HIV DNA vaccine [29]. For this purpose, various approaches have been proposed for mannose incorporation on the surface of LNPs [14]. Recently, we have reported a method for mannosylation of LNP by a stable Man-cholesterol amine conjugate, showing that the novel delivery systems improved the potency of an influenza self-amplifying mRNA (SAM) vaccine [30]. SAM promotes RNA amplification and requires lower dose compared to conventional mRNA vaccines [31,32].

Due to recent interest in liposomal delivery of mRNA vaccines, our group undertook the task of exploring a way to further improve our previously described LNP with mannans of length higher than the monosaccharide. In particular, among a set of 275 glycans, oligomannans (Figure 1) have been assessed by microarray to be strong ligands to DC-SIGN [33], therefore were selected for this study.

For this investigation, we chose respiratory syncytial virus (RSV) F protein as a model antigen. RSV is a common respiratory mRNA virus often causing mild, cold-like symptoms [34]. RSV is the most common cause of bronchiolitis and pneumonia in children younger than 1 year of age in the United States and is also a significant cause of respiratory illness in older adults. Despite the medical needs and constant preclinical and clinical efforts over the last 5 decades, a vaccine is not available yet. The only preventive measure available is based on by immune-prophylaxis with the RSV neutralizing monoclonal antibody Palivizumab, which targets the RSV F protein [35], a surface glycoprotein responsible for viral fusion to the host cells. Therefore, inducing neutralizing antibodies against RSV F protein has been the primary focus of the investigational vaccines [36,37]. In spite of these encouraging premises, failure of a recent Phase 2b clinical trial by Novavax investigating a RSV F-based nanoparticle vaccine candidate indicated the challenges associated with the development of a vaccine to prevent RSV [35] and the need of improved formulations [38,39].

Herein, we constructed LNPs exposing mannans, varying from mono- to tetrasaccharide, and explored the impact of the carbohydrate length on the antibody response elicited of an RSV-F protein-based SAM vaccine, comparing the intramuscular and intradermal route.

## 2. Materials and Methods

### 2.1. Mannan-Cholesterol Conjugation

Structures **M1**–**M4** were synthesized with a propylamine linker for conjugation as reported in literature [40]. For this purpose, a 4-(*p*-nitrophenol)oxobutanamide group at the position 3 of cholesterol molecule was incorporated by reaction of (3β)-Cholest-5-en-3-amine or cholesterol amine with succinin anhydride and following conversion of the formed acid into *p*-nitrophenyl (Scheme 1) [30]. In a typical conjugation experiment, commercial (3β)-cholest-5-en-3-amine 5 (77 mg, 0.2 mmol) in CH_2_Cl_2_ (2 mL) was treated with 2 equiv of succinic anhydride. The intermediate acid was reacted with EDC (4 equiv) and *p*-nitrophenol (4 equiv) and the crude mixture was purified with a gradient cyclohexane → EtOAc on a Redisep silica gel column using a Combiflash instrument, to yield the pre-activated cholesterol amine 6 (60%) (Appendix A). The intermediate 6 was reacted with the mannans 2–4 (0.1 mmol) in DMSO (1 mL) containing triethylamine (50 µL), monitoring by TLC (dichloromethane–methanol). After stirring overnight, the conjugate was precipitated by the addition of 18 mL of EtOAc and dried to obtain the desired conjugates **M2**–**M4** (55–75%). After confirming the structure by ^1^H-NMR (Appendix A), the conjugates were used for LNPs formulation.

### 2.2. Preparation of LNPs and Surface Decorated LNPs

Classical cationic LNPs were prepared as described by Geall and coworkers [31], using the molar ratios of 40% DLin-DMA, 10% DSPC, 48% cholesterol, and 2% DMG-PEG (2000). Briefly, the lipids 1,2-dilinoleyl oxy-3-dimethylaminopropane (DLinDMA),51 1,2-distearoyl-sn-glycero-3-phosphocholine (DSPC; Genzyme, Cambridge, MA, USA), 1,2-Dimyristoyl-sn-glycero-3-phosphoethanolamine-*N*-[methoxy(polyethylene glycol)-2000] (ammonium salt) (PEG-DMG 2000; Avanti Polar Lipids, Alabaster, AL, USA) and cholesterol (Sigma-Aldrich, St Louis, MO, USA) were used. The formulations were prepared at nitrogen to phosphate ratio of 8:1. 

For the preparation of MLNPs, the mannan conjugated cholesterol was added by partly replacing the cholesterol content of the formulation. The formulations were prepared using the ethanol injection method using the T junction, which mixes 1 stream of lipids dissolved in ethanol, 1 stream of mRNA dissolved in 100 mM citrate buffer (pH 6), and 1 stream of only citrate buffer (same strength), at a constant flow rate (all solutions preheated to 37 °C before loading to the T junction). The formulations were later buffer exchanged with 100 mM phosphate buffer using PD 10 column. MLNP formulation optimization was done by preparing MLNPs with varying concentrations of mannan-cholesterol conjugates and PEG, buffer variation was also done whenever necessary.

### 2.3. Particle Size Measurement

Particle size was measured using the Malvern Zetasizer instrument (Malvern Spectris, Egan, England) using 1:500 dilution of the LNP formulations in PBS buffer. The particle size data also gave a polydispersity index (PDI). The analysis was done first to characterize the particles with different concentration of mannans and then for particles with varying PEG content.

### 2.4. Concanavalin A (ConA) Binding Assay

ConA induced LNP aggregation was carried out by incubating 10 µL of LNPs to 1.5 mL of 0.1 M HEPES buffer containing ConA at a concentration of 100 µg/mL and 1 mM of Ca^2+^ and Mn^2+^ salts. The increase in size due to ConA binding to mannans was measured using Malvern Zetasizer after specific intervals. The particles were per-incubated in 1 mg/mL BSA solution for 6 h in order to de-PEGylate them whenever necessary.

### 2.5. Ribogreen Assay for RNA Quantification

Triton X-100 breaks the LNPs apart and releases the RNA entrapped inside the LNPs which when quantified gives total RNA in the formulation, whereas quantifying RNA without Triton evaluates the RNA bound to the surface of LNPs. The difference between the two readings gives the amount encapsulated by the LNPs. Formulation samples were serially diluted in TE buffer with and without 0.1% Triton X-100. Then, the samples were incubated with Ribogreen reagent diluted 200-fold using TE buffer at 37 °C for 15 min. Fluorescence was measured after 10 min at 485 nm. This method was used to quantify the SAM antigen encapsulated by the LNPs. RSV SAM was used as model antigens to study the encapsulation efficiency of the MLNPs.

### 2.6. Gel Electrophoresis

Gel electrophoresis was performed on the formulation samples using 1% agarose gel. Phenol chloroform extraction assay was performed on the formulation samples to extract the RNA from the LNP formulations prior to loading the samples. SYBR gold dye staining was used to visualize the RNA in the gel under UV light using Gel Doc. Same model antigens were used to study the SAM antigen integrity after encapsulation inside MLNPs.

### 2.7. Immunizations

Protocols were approved by the Italian Ministry of Health. All mice were housed under specific pathogen-free conditions at the GSK Vaccines Animal Resource Center. Five animals per group of 6 to 8 weeks old female BALB/c mice, obtained from Charles River, were used in the study. Animals were immunized IM or ID, with two 0.1 µg doses, 8 weeks apart, M1LNP, M2LNP, M3LNP, and M4LNP adjuvanted vaccine as test groups. LNP adjuvanted vaccine (standard formulation) and non-adjuvanted RSV F subunit protein vaccine (0.1 µg dose) were the controls. The volume of injection was 20 µL for both IM and ID immunizations. Sera samples were collected after 2 weeks of each immunization. 

### 2.8. ELISA Titers

Determination of anti-RSV specific total IgG and subclasses IgG1 and IgG2a titers was performed by ELISA using individual serum samples. Maxisorp plates (Nunc) were coated overnight at +4 °C with 100 µL/well of 2 µg/mL solution of RSV subunit solution in pH 8.2 PBS buffer, and blocked for 1 h at 37 °C with 200 µL of 1% BSA (Sigma Aldrich, Saint Louis, MO, USA), in PBS. For the IgG ELISA assay, serum samples were diluted appropriately using 0.1% BSA and 0.05% Tween-20 (Sigma Aldrich, Saint Louis, MO, USA) and transferred into coated and blocked plates and serially diluted (1:2) and incubated at 37 °C for 2 h. Bound RSV-specific IgG, IgG1, and IgG2a were detected by incubating with alkaline phosphatase-conjugated goat anti-mouse IgG, IgG1, and IgG2a solution in PBS, respectively (SouthernBiotech, Birmingha, AL, USA). Subsequently, the plates were washed and incubated with 100 µg/well of 4-nitrophenyl phosphate (Sigma Aldrich, Saint Louis, MO, USA) solution in diethanolamine reagent. Antibody titers, normalized respect to the reference serum assayed in parallel, were obtained using IrisLab software (Chiron, Siena, Italy).

GraphPad Prism v.6.04 (GraphPad Software, La Jolla, CA, USA) was used to graph and analyze data for statistical significance. Intergroup comparison was analyzed using the Mann–Whitney test.

## 3. Results

### 3.1. Preparation of Mannosylated LNP

The LNP formulation used in the current work (Scheme 1A) is composed of ionizable amino lipid (DLin-DMA), a zwitterionic phosphatidylcholine (1,2-distearoyl-sn-glycero-3-phosphocholine, DSPC), cholesterol and a coat lipid (polyethylene glycol-dimyristolglycerol, PEG-DMG) at a fixed molar ratio of 40:10:48:2 [30]. The head-group of DLin-DMA lipids contains tertiary amines with a pKa 6–7 which can be readily protonated at lower pH and become cationic in the endosomes, facilitating endosomal escape LNPs to cytosol. LNPs with neutral charge at physiological pH appear to be less immunoreactive than cationic lipids with a fixed charge such as DOTAP [10,11]. DSPC interacts with the mRNA through its choline group and helps stabilizing the LNP both during formulation and circulation. Cholesterol gets evenly dispersed on the surface and core, ensuring efficient lipid packaging and providing rigidity to the particles. PEG coating enables to control particle size during formulation, preventing aggregation and enhancing the stability of particles. LNPs herein described were designed to shed PEG rapidly upon intravenous injection [11], thus exposing the mannoses in vivo.

For mannose incorporation onto LNP, part of cholesterol was replaced with a hydrolytically stable conjugate of mannose glycans with cholesterol amine, as we described recently [30]. We also envisaged that the larger was the mannan, the lower would be the masking effect from PEG, so that receptor engagement would occur also in case of delayed PEG shedding.

To achieve the synthesis of the target conjugates, the mannan structures **1**–**4** were prepared with an aminopropyl linker as we recently reported in literature [40]. Mannose **1** was conjugated to cholesterol as described in our recent report [30].

To obtain conjugates of mannans **2**–**4**, cholesterol amine was transformed into *p*-nitrophenol ester, as shown in Scheme 1, for following condensation with the glycans to provide the final conjugates **M2**–**M4**.

### 3.2. Formulation Optimization

Based on our previous work, the molar concentration of mannosylated cholesterol to be incorporated to the MLNPs was targeted to be 15% [30], in order to maintain the particle integrity. Accordingly, different relative percentages of cholesterol and PEG were tested as reported in Table 1.

Reduction of PEG content was attempted to rule out any potential ligand masking effect of PEG chains. As shown in Table 2, physical characteristics of the formulations revealed that 2% and 0.3% PEG formulations presented acceptable size (<200 nm) and PDI (<0.2) values, also in the presence of higher chain length mannans. In addition, 0.3% PEG formulations were found to have an overall greater size than the 2% PEG formulations, indicating the reduction in PEG leads to higher particle size irrespective of the mannan used. Size (>200 nm) and PDI (>0.2) were behind the target when PEG was not used in the formulation. Henceforth, 2% PEG (as in classical LNPs) and 0.3% PEG formulations were considered for further studies. 

The developed LNPs were used for encapsulation of the RSV F protein antigen encoded as SAM vaccine. The encapsulation efficiency was measured through a Ribogreen assay. The results showed greater than 70% SAM encapsulation occurring in all formulations (Table 2).

Then, antigenic integrity was then checked using agarose gel electrophoresis. All formulation samples were treated with ethanol and sodium acetate to extract RNA and loaded on agarose gel. The result confirmed the antigenic integrity of the SAM LPNs as compared to the unformulated drug substance standard (Figure 2A).

In order to assess the proper exposition of the mannans of the LNP surface, the formulations were tested with ConA binding assay (Figure 2B). ConA is known to preferentially binds mannose ligands, so the particles were incubated with the lectin, the variation of the particle size after incubation was monitored [30]. The experiment showed that even with 2% PEG formulations, MLNPs were recognized by ConA. The M3-LNP-2% PEG formulation showed higher recognition compared to the shorter **M1** and **M2** ligands, as expected for the stronger binding to the lectin. Unexpectedly, **M4** did not show better recognition than **M1** and **M2**, indicating that possibly the additional branching in the structure were diminishing the ligand interaction. The recognition of the four ligands **M1**–**M4** was found to greatly enhance for all 0.3% PEG formulations. Therefore, PEG reduction in MLNPs favors ligand recognition on its surface in vitro. However, the mannans with higher chain length could be recognized even at 2% PEG concentration on MLNPs.

### 3.3. In Vivo Evaluation

Initially, the 2% PEG MLNP formulations were tested for their delivery and adjuvanticity of alpha virus-based SAM replicon expressing RSV F protein through the conventional IM route. Groups of five 6 to 8-week-old female BALB/c mice received two 0.1 µg doses, 8 weeks apart. The subunit vaccine was the positive control.

IgG titers were measured by ELISA after the second dose an improved immune response in the presence of the mannosylated LNP compared to the un-glycosylated form which became higher at the increase of the length.

Particularly, M4LNP was found to enhance the IgG titers for SAM vaccine when compared to LNP and both **M1** and M2LNP at the second dose, whereas the response among the other mannosylated formulations was found to be comparable (Figure 3A).

Interestingly, while IgG2a titers were comparable among all the LNP formulations, regardless of mannosylation, a significant improvement in IgG1 titers was obtained at onset of immune response with the M1LNP and M2LNP adjuvanted SAM vaccine as compared to the SAM LNP vaccine and subunit control vaccine. However, the response plateaued thereafter as well as among all groups after second immunization. SAM vaccines enhanced the IgG titers and IgG2a with greater intensity after single immunization, which was comparable with that obtained after 2 immunizations with the subunit control vaccine.

After assessing the potential of the mannosylated via IM, the effect on the immunogenicity via ID route, where the uptake by Langerhans cells is expected to be favored [30], was explored. Two in vivo studies were undertaken using RSV SAM to compare the 2% and 0.3% PEG MLNP formulations, respectively, with the plain LNP. Again, groups of five 6 to 8-week-old female BALB/c mice received two 0.1 µg doses, 8 weeks apart. The subunit vaccine was the positive control. Sera were sampled 2 weeks after first and second immunization and tested for total IgG and IgG subclasses (IgG1 and IgG2a). 

In the 2% PEG MLNP formulations, after one dose, the improvement of titers obtained with the different SAM MLNP formulations appeared marginal, if groups are compared to each other, since differences were not statistically meaningful (Figure 4A). However, a continuous trend of increase in the total IgG titers with increasing chain length of mannan (from **M1** to **M4**) formulations could be clearly detectable. Notably, **M4** led to a significant enhancement of the IgG response when compared to the subunit vaccine. A trend of increase in the IgG1 titers following first immunization with the increasing chain of mannans in MLNPs (2% PEG formulations), could be observed, albeit differences were not statistically significant (Figure 4B).

After the second immunization, however, SAM LNP per se was found to remarkably enhance the IgG titers as compared to subunit control, thus masking the effect of mannosylation. 

Only M1LNP showed an improved immunogenicity compared to plain LNP. Interestingly, in general IgG1 levels induced by SAM LNP and mannosylated formulations did not show remarkable differences (Figure 4B), while augmented IgG2a levels at the increase of the mannan length could be observed (Figure 4C) compared to subunit control, both after the first and second dose. 

When PEG content was lowered to 0.3%, formulations adjuvanted with M2LNP, M3LNP, and M4LNP confirmed to lead to increased IgG titers in comparison to the subunit vaccine (Figure 5A).

As observed with 2% PEG, a trend of increased response was measurable for both IgG1 and IG2a titers elicited by the 0.3% PEG LNP formulations (Figure 5B). Particularly, after the first immunization, the IgG2a titers induced by M2LNP, M3LNP, and M4LNP formulations were higher than LNP adjuvanted SAM vaccine. However, after the second immunization, only the M2LNP formulation resulted in titers higher than those of the conventional LNP formulation, while titers measured in the other MLNP immunized groups were found to be comparable (Figure 5C). 

Noteworthy, while M2–M4LNP formulated with 2% PEG MLNP were found to have slighlty lower antibody titers (both in terms of total IgG and IgG1 or 2a subclasses) than conventional LNP and M1LNP, this reduction of titers was not observed in 0.3% PEG formulations. This could result from a better mannan ligand display on the liposomal surface with a lower PEG incorporation. 

Overall, considering the titers at the priming dose, there was a clear trend of increase in the titers with increase in the mannan chain length, irrespective of the PEG content of the formulations. 

## 4. Discussion and Conclusions

Mannans are reported to bind CRD on the C-type lectin receptors of the APCs specifically mannose receptor, DC-SIGN receptor, and langerin receptor [41]. Mannose and oligomannas are known to be optimal ligands for DC-SIGN [33]. 

Previous studies have underlined that enhanced uptake of mannosylated delivery systems by immature DCs and macrophages results in enhanced immune response to vaccines [13,42]. We have recently demonstrated that a mannose-cholesterol conjugate incorporated in LNPs in vitro enhanced uptake of the particles in comparison to the un-glycosylated counterpart from bone marrow-derived dendritic cells, and in vivo induces a more rapid onset of the antibody response, of an influenza SAM vaccine [30]. 

Herein, the in vivo effect of the mannans length was explored by preparing mannan-cholesterol conjugates from a series of ligands with different chain length, namely Man monomer (**M1**), dimer (**M2**), trimer (**M3**), and tetramer (**M4**). The mannan cholesterol conjugates were incorporated to an extent of 15% of the cholesterol amount used in the LNP formulation. The produced MLNPs were characterized for their physical characteristics, including particle size, homogeneity, encapsulation efficiency, and antigenic integrity.

The surface presentation of the mannans was then tested by the binding with the ConA lectin [43]. Higher binding to ConA was observed with higher mannan length at 2% PEG content, with the best results obtained from M3LNP-2% formulation. Reduction of the PEG content from 2% to 0.3% did not compromise the physical attributes of the MLNPs but increased the exposure of all the mannans. 

The novel MLNPs were tested in vivo using RSV SAM as the model antigen. Initial data with 2% PEG MLNP formulations through IM route showed a strong onset of immune response with MLNP adjuvanted RSV SAM vaccine which was comparable to that elicited by two doses of the non-adjuvanted subunit vaccine. This finding could have a high significance towards the development of a single dose vaccine. 

By IM, the mannosylation of LNP seemed to act at the first dose increasing particularly in increasing the IgG1 isotypes, specifically with **M2** compared to **M1**. The IgG2a levels were comparable in spite of the mannans length and higher than the subunit vaccine, highlighting the potential of LPN per se in augmenting this antibody subclass. 

When administered via ID route, after first immunization, both 2% and 0.3% PEG MLNPs lead to a trend of increased IgG titers at the growth of the mannan length over SAM LNP and subunit vaccines. Such a trend was not observed at the booster dose, presumably because of saturation of the immune response. The separation of the administered doses in this study was based on typical schedules established for conventional vaccines. This might have impacted the booster dose response of the mRNA vaccine and optimization of time priodi between prime and boost dose for such vaccines could further optimize the observed immune response. 

The differences in response were clearly visible in the IgG subclass titers. M1LNP was confirmed to be the best adjuvant for enhancing the IgG1 subclass titers irrespective of PEG content of MNPs formulations after 2 doses. However, further enhancement of the immune response was seen with M2LNP in the 0.3% PEG formulations, while the IgG1 titers plateaued above this length. MLNPs particularly with 0.3% PEG formulations appeared to increase the IgG2a titers compared to LNP. M2LNP with 0.3% PEG exhibited higher IgG2a titers than the un-glycosylated LNP SAM vaccine after 2 immunizations. Notably, the IgG2a response obtained after 1 immunization with MLNPs was comparable to that obtained after 2 doses of the subunit vaccine, indicating the strong adjuvant effect of this formulations [44]. Therefore, while in general, MLNPs showed to induce IgG1 antibodies, indicative of Th2 biased response, reduction of PEGylation improved the exposure of mannans inducing a more robust Th1 response. 

In conclusion, our results indicate that mannans longer than the monosaccharide were helpful in enhancing the immunogenicity of the SAM-MLNP vaccine by intramuscular and intradermal route. In general, the increase in the response plateaued when a size larger than a disaccharide were used. 

Therefore, incorporation of a mannose disaccharide ligand on the LNP surface can augment the immunogenicity of SAM vaccines compared to conventional LNPs, while no additional benefit is associated with incorporation of longer mannans. In addition, reduction of PEG content proved important to ameliorate the SAM LNP immunogenicity, possibly because of the better ligand exposure. These findings highlight that LNP mannosylation can be very instrumental in inducing required immune response to the vaccine upon a single dose, an attribute highly desirable in developing single shot vaccines for pandemic preparedness. Additionally, these data support the use of the intradermal route, with potential in self-administered dermal patches development for prolonged release and minimization of healthcare workers intervention for vaccine administration. 

## Data Availability

The authors declare that the data supporting the findings of this study are available within the paper and its Appendix A files.

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
