# Peer review of "Conjugation of Mannans to Enhance the Potency of Liposome Nanoparticles for the Delivery of RNA Vaccines"

_pharmaceutics, 2021, doi:10.3390/pharmaceutics13020240_

Round 1

Reviewer 1 Report

The Article entitled "Conjugation of mannans to enhance the potency of LNP for the delivery of RNA vaccines" is very well written and presented. The researchers generated sets of LNPs with a modified surface with mannans of different lengths (from mono to tetrasaccharide) to study the effect on the SAM model replicon encoding's antibody response the respiratory syncytial virus fusion protein. Besides, they were shown the impact of mannosylated liposomal delivery by intradermal and intramuscular routes. My suggestion is to accept with a minor review, please per attention to some details: 1. figures 3, 4 and 5 - improve the quality of axis x. 2. linea 107: there are two dots. 3. Table 2: show the STDEV of Entrapment (%).

Author Response

STDEV added in Table 2 

Reviewer 2 Report

In this work, Baudner, Adamo and coworkers study the effect of mannose on the antibody production efficiency of Self Amplifying RNA (SAM) vaccine. The authors found that mannose disaccharide ligand on lipid nanoparticle (LNP) surface can enhance the immunogenicity of SAM vaccines compared to conventional LNPs. Incorporation of longer mannans does not increase antibody production efficiency. Importantly, mannosylated LNPs have a comparable immune response to that elicited by two doses of the non-adjuvanted subunit vaccine. As such, this work is highly important towards the development of a single-dose vaccine. Overall, this work is novel, well-designed, well-conducted and well-presented. I strongly recommend publishing this manuscript on Pharmaceutics after revisions. I have several comments below to improve the clarity of the manuscript. None of these comments affects the already high quality of the work.

  1. In the abstract, the authors report that “This work confirms the potential of mannosylated SAM LNPs for intradermal delivery…”. However, in the conclusion, the authors wrote that “our results indicate that mannans longer than the monosaccharide were helpful in enhancing the immunogenicity of the SAM-MLNP vaccine by intramuscular and intradermal route.” I am not sure why the intramuscular route was left out in the abstract.
  2. The sentence “Muscle cells have been shown involved in presentation of RNA and DNA vector encoded antigens has been suggested” should cite a recent relevant work (doi.org/10.3390/pharmaceutics12111068).
  3. In the introduction, the authors highlight that “Carbohydrate-recognition domain (CRD) on such calcium-dependent C-type lectin receptors of DC-SIGN and Langerin are optimal for targeting using mannosylated-LNP for the delivery of SAM vaccines through the intradermal route instead of conventional intramuscular or subcutaneous routes for vaccines.” It would be helpful if the authors discuss this point further. Why is intradermal route instead of conventional intramuscular or subcutaneous routes optimal for targeting using mannosylated-LNP? The data reported in this work seem to show that the intramuscular route is also comparable to the intradermal route so the authors may discuss this point in more details.
  4. On page 12 and 13, it is discussed that “LNPs herein described were designed to shed PEG rapidly upon intravenous injection[11], thus exposing the mannoses in vivo.” However, this work only studies intramuscular and intradermal injection, so does the PEG also shed off the LNP surface in this study?
  5. The authors also envisaged that the larger was the mannan, the lower would be the masking effect from PEG. This point seems to contradict the statement that if the PEG detaches from the LNPs surface in-vivo, PEG should not affect the mannose interaction with APCs receptors.
  6. Line 229, “PDI.3.1” may be a typo.
  7. The entrapment % reported in Table 2 may have too many digits. 72% may be written instead of 71.78.
  8. In Figure 3, 4, and 5, ** p < 0.01, *** p < 0.001 are noted in the caption but not in the figures. It is unclear which data has this level of significance.

Author Response

  1. In the abstract, the authors report that “This work confirms the potential of mannosylated SAM LNPs for intradermal delivery…”. However, in the conclusion, the authors wrote that “our results indicate that mannans longer than the monosaccharide were helpful in enhancing the immunogenicity of the SAM-MLNP vaccine by intramuscular and intradermal route.” I am not sure why the intramuscular route was left out in the abstract.

Response: Modified accordingly.

  1. The sentence “Muscle cells have been shown involved in presentation of RNA and DNA vector encoded antigens has been suggested” should cite a recent relevant work (doi.org/10.3390/pharmaceutics12111068).

Response: Added

  1. In the introduction, the authors highlight that “Carbohydrate-recognition domain (CRD) on such calcium-dependent C-type lectin receptors of DC-SIGN and Langerin are optimal for targeting using mannosylated-LNP for the delivery of SAM vaccines through the intradermal route instead of conventional intramuscular or subcutaneous routes for vaccines.” It would be helpful if the authors discuss this point further. Why is intradermal route instead of conventional intramuscular or subcutaneous routes optimal for targeting using mannosylated-LNP? The data reported in this work seem to show that the intramuscular route is also comparable to the intradermal route so the authors may discuss this point in more details.

Response: DC-SIGN receptors are found on immature DCs of muscle as well, which can enable targeting to intramuscular DCs. Explanation given with reference.

  1. On page 12 and 13, it is discussed that “LNPs herein described were designed to shed PEG rapidly upon intravenous injection[1], thus exposing the mannoses in vivo.” However, this work only studies intramuscular and intradermal injection, so does the PEG also shed off the LNP surface in this study?

Response: The above reference is made to Material and methods section in page 6, where classical LNP preparation is described. Classical LNPs have a tendency to shed PEG upon intravenous injection. This fact was emphasized to justify selection of Cholesterol for ligand conjugation, because with PEG chains on surface, the ligands could be masked. We have also demonstrated it through in-vitro experiment of Con A binding assay (Fig 2 B), where MLNP formulations with higher mannans as well as with reduced PEG (0.3% vs 2%) have shown better ligand recognition. Our group has studied and published previously  [ ACS Infect Dis, 2019. 5(9): p. 1546-1558], the same assay with depegylation of LNPs using BSA incubation to mimic serum proteins and found that depegylated LNPs were recognizing the ligands better over a period of time than depegylated LNPs. That’s why we do not intend to publish similar results in this paper. However, in-vivo investigation of PEG shedding was not performed during the study and we recommend such testing in future experiments.

  1. The authors also envisaged that the larger was the mannan, the lower would be the masking effect from PEG. This point seems to contradict the statement that if the PEG detaches from the LNPs surface in-vivo, PEG should not affect the mannose interaction with APCs receptors.

Response: We are very thankful to the reviewers for reviewing the manuscript with details of such fine extent. PEG shedding usually does not happen immediately and follows its own kinetics to reach complete depegylation in-vivo [ Mol Pharm, 2015. 12(2): p. 386-92]. Considering this fact, as the vaccine is delivered to the nearest vicinity of its target (APC are readily accessible upon IM or ID delivery), there were fair chances that MLNPs were internalized by other cell population (like muscle cells) if ligands are not exposed due to delayed PEG shedding. This is the reason why the authors envisaged to test the impact of higher mannans, so as to ensure better ligand exposition even in the case of delayed PEG shedding.

We have better clarified this in ln 217-218 pg 6.

  1. Line 229, “PDI.3.1” may be a typo.

Response: Corrected

  1. The entrapment % reported in Table 2 may have too many digits. 72% may be written instead of 71.78.

Response: We usually report to 2 places of decimal for accuracy. Corrected, however.

  1. In Figure 3, 4, and 5, ** p < 0.01, *** p < 0.001 are noted in the caption but not in the figures. It is unclear which data has this level of significance.

Response: Corrected

Reviewer 3 Report

To the authors, I hope this message finds you well. The current manuscript is of high quality. A nanoparticle-based COVID-19 vaccine may be cheap, safe, and effective. Development of a novel nanoplatform for generation of new vaccines is the main question that has been addressed in this study. I believe that the authors could successfully hypothesize and prove their ideas by meticulously designed experiments. However, there are some information that should be provided for the readers: 1- How did you evaluate the mannan-cholesterol conjugation. The reader needs to be provided by an evidence such as NMR or Mass Spec data for this conjugation. 2- A schematic representation of nanoparticle production would help the readers to better understand the method. Regards

Author Response

To the authors, I hope this message finds you well. The current manuscript is of high quality. A nanoparticle-based COVID-19 vaccine may be cheap, safe, and effective. Development of a novel nanoplatform for generation of new vaccines is the main question that has been addressed in this study. I believe that the authors could successfully hypothesize and prove their ideas by meticulously designed experiments. However, there are some information that should be provided for the readers: 1- How did you evaluate the mannan-cholesterol conjugation. The reader needs to be provided by an evidence such as NMR or Mass Spec data for this conjugation. 2- A schematic representation of nanoparticle production would help the readers to better understand the method. Regards

Response: Scheme 1 describing chemical synthesis of Mannan-cholesterol conjugates has been added. Paragraph 2.1, describing the preparation of the conjugates, was modified accordingly. NMR spectra are now provided as Supplementary Material.

Reviewer 4 Report

Goswami et al. studied the “Conjugation of mannans to enhance the potency of LNP for the delivery of RNA vaccines”.

In the present study, authors have prepared the novel mannan(s) decorated LNPs. To determine the efficacy of the MLNPs, authors have used Respiratory Syncytial Virus Fusion F protein as a model antigen, and also evaluated the same compositions in the different route of immunizations. Results revealed that mannan decorated LNPs are more efficacious vaccine adjuvant candidates to un-glycosylated LNPs. In addition, authors have established the precise structural-activity relationship of MLNPs against RSV-F protein. Data is well presented. Discussion is well placed.

I have a few suggestions and queries.

In the title: Please expand the “LNP”, which will be easy for search engines to showcase best hits.

Although discussion is well placed about the MLNPs as novel adjuvants, a brief description of mannan-conjugates as adjuvants and/or immunomodulators is missing.  

This reference is useful for a short description and citable:   https://doi.org/10.1016/j.tips.2017.06.002

Line 82-85: https://doi.org/10.1016/j.intimp.2019.105684. This reference deserves citation here

Line 158: delete “solution in TE buffer”.

Line 168: “antigens were to study” -------------“antigens were used to study”

Figure 2A: Scale of the ladder is missing.

For all the figure, please try to mention the details like n=?, SD or SEM, and others.

In figure 3, MLNPs are shown to induce IgG1 response, which is indicative of Th2 biased response. However, in figure 5 with 0.3% PEG, the results are Th1 biased. Thus, less PEGylation and exposure of mannans induces robust Th1 response. Please place the discussion.   

Figure 4B: How authors justify the decreased antibody response (IgG1) while increasing mannan length.

Authors would have been evaluated for Th1/Th2 cytokine secretion either in vivo or ex vivo (as recall response). Is there any specific reason to miss over this part?

Author Response

In the title: Please expand the “LNP”, which will be easy for search engines to showcase best hits.

Response: Corrected

Although discussion is well placed about the MLNPs as novel adjuvants, a brief description of mannan-conjugates as adjuvants and/or immunomodulators is missing.

Response: Thanks for this suggestion. Though mannosylation of nanoparticles for DC SIGN targeting has been primarily attempted for some therapeutic indications using modified LNPs for delivering the sugar antigens (as discussed in the introduction), there are not many reported examples where mannan conjugates have been used as adjuvants or immunomodulators. We found a few references (one is suggested the review below) that we are now quoting (https://doi.org/10.1016/j.tips.2017.06.002 and https://doi.org/10.1016/j.intimp.2019.105684) in pg 2 ln 65-76.

Line 82-85:. This reference https://doi.org/10.1016/j.intimp.2019.105684 deserves citation here

Response: Included, see above.

Line 158: delete “solution in TE buffer”.

Response: Edited

Line 168: “antigens were to study” -------------“antigens were used to study”

Response: Edited

Figure 2A: Scale of the ladder is missing.

Response: Scale information included in figure caption

For all the figure, please try to mention the details like n=?, SD or SEM, and others.

Response: Corrected

In figure 3, MLNPs are shown to induce IgG1 response, which is indicative of Th2 biased response. However, in figure 5 with 0.3% PEG, the results are Th1 biased. Thus, less PEGylation and exposure of mannans induces robust Th1 response. Please place the discussion.

Response: Thanks for highlighting this, we added these considerations in the discussion, pg 12 ln 399.

Figure 4B: How authors justify the decreased antibody response (IgG1) while increasing mannan length.

Response: The visual presentation of data indicates increase in IgG1 titers after the prime dose, however, a reduction is visible after the booster dose. Though with 8 mice per group, we did not find the differences statistically meaningful. That’s why not discussed in detail. ID route has been investigated for many conventional vaccine and it has demonstrated significant dose sparing potential [ Bull World Health Organ, 2011. 89(3): p. 221-6.], which are in accordance with the results obtained (Higher titers upon ID immunization post 1 as well as post 2) reduction in IgG1 response with MLNPs. We separated the 2 doses based on conventional approach of 28 days apart. The separation of doses need further investigation for mRNA based vaccines (to mimic the same increase in titers post 2 dose, as observed after the single dose) for a more optimized response after the booster dose. Added in discussion pg 12 ln 383 on.

Authors would have been evaluated for Th1/Th2 cytokine secretion either in vivo or ex vivo (as recall response). Is there any specific reason to miss over this part?

Response: The Th1/Th2 cytokine secretion profile was explored in the previous work, which has utilized only mono-saccharide as ligand [ACS Infect Dis, 2019. 5(9): p. 1546-1558]. Current study has utilized higher mannans, so a lot of effort was made on synthesis and formulation optimization. Unfortunately, the cytokine profile analysis was not undertaken due to limited time of the PhD program, and it was challenging to repeat animal immunization incorporating this read out during the covid-19 time. However, we will pursue this investigation in future studies.